# Exploring Public Preferences, Priorities, and Policy Perspectives for Controlling Invasive Mosquito Species in Greece

**DOI:** 10.3390/tropicalmed4020083

**Published:** 2019-05-18

**Authors:** Antonios Kolimenakis, Dionysios Latinopoulos, Kostas Bithas, Clive Richardson, Konstantinos Lagouvardos, Angeliki Stefopoulou, Dimitrios Papachristos, Antonios Michaelakis

**Affiliations:** 1Research University Institute of Urban Environment and Human Resources, Panteion University, Athens 17671, Greece; akolimenakis@gmail.com (A.K.); kbithas@gmail.com (K.B.); crichard@panteion.gr (C.R.); 2School of Spatial Planning and Development, Aristotle University of Thessaloniki, Thessaloniki 54124, Greece; 3National Observatory of Athens/Institute for Environmental Research, Athens 15236, Greece; lagouvar@noa.gr; 4Department of Entomology and Agricultural Zoology, Benaki Phytopathological Institute, Kifissia-Athens 14561, Greece; a.stefopoulou@bpi.gr (A.S.); d.papachristos@bpi.gr (D.P.); a.michaelakis@bpi.gr (A.M.)

**Keywords:** urban ecosystems, climate change, Asian tiger mosquito, web survey, infectious diseases, citizens’ perception

## Abstract

Climate change, urbanization, and financial crisis have created a dramatic mixture of challenges in Southern Europe, increasing further the risks of transmission of new vector-borne diseases. In the last decade, there has been a wide spread of an invasive mosquito species *Aedes albopictus,* commonly known as the Asian tiger mosquito, in various urban ecosystems of Greece accompanied by greater risks of infectious diseases, higher nuisance levels, and increased expenses incurred for their control. The aim of the present paper is to investigate citizens’ perception of the *Aedes albopictus* problem and to evaluate various policy aspects related to its control. Findings are based on the conduct of a web-based survey at a national scale and the production of national surveillance maps. Results indicate that citizens are highly concerned with the health risks associated with the new mosquito species and consider public prevention strategies highly important for the confrontation of the problem while, at the same time, surveillance maps indicate a constant intensification of the problem. The spatial patterns of these results are further investigated aiming to define areas (regions) with different: (a) Levels of risk and/or (b) policy priorities. It appears that citizens are aware of the invasive mosquito problem and appear prone to act against possible consequences. Climate change and the complex socio-ecological context of South Europe are expected to favor a deterioration of the problem and an increasing risk of the transmission of new diseases, posing, in this respect, new challenges for policy makers and citizens.

## 1. Introduction

According to WHO (2017) [1], many countries are still unprepared to address the looming challenges of vector-borne diseases, which are further intensified by the strong influence of social and environmental factors on vector-borne pathogen transmission. Therefore, a critical necessity arises for an informed restructuring of national control and surveillance programmes in order to address the risks posed by multiple vectors and diseases as well as a high preparedness level of national health systems. All these challenges require an increased level of information with regards to the effectiveness of control interventions, well-trained specialised staff who can build sustainable systems for their delivery and a high level of citizens’ awareness necessary for the control of *Aedes* species [1].

In recent years, concern has arisen over the threats of an increase in mosquito-borne diseases in the Mediterranean as new sanitary and environmental risks are emerging, including the appearance of chikungunya (CHIK) and reappearance of dengue (DENV) and West Nile (WNV) viruses, requiring the adoption of specific measures and strategies by both policy makers and scientists. These vector-borne diseases (VBDs) in Europe are associated with the presence of the invasive mosquito species (IMS) such as the Asian tiger mosquito (*Aedes albopictus*) and *Aedes aegypti*. In Europe, the only IMS that is present is *Ae. albopictus,* except the Madeira islands (Portugal) where *Ae. aegypti* is established [2,3]. The first presence of *Ae. albopictus* in Greece (in Northwest prefectures) is dated back to 2003 [4], while in Athens (Attica Region), it was confirmed for the first time in 2008 [5]. The mosquito-associated problem in Greece, as also in other parts of Europe, has been recently intensified and favored by both geographic position and climatic conditions of Greece. It should be noted that Greece is a representative case of Mediterranean climate. During spring and summer, the major part of the country experiences small rainfall amounts, with the exception of the mountainous areas of Western and Northern Greece where thunderstorm activity is frequent [6]. During autumn and winter, rainfall is more abundant over the western part of continental Greece as well as over the western part of Crete island (with yearly accumulations up to 2000 mm), while the eastern part of continental Greece as well as the islands of the Aegean Sea are much drier (with yearly accumulations up to 400–600 mm). During summer, the eastern part of the country as well as the Aegean Sea are influenced by strong northern winds, named etesians [7], a wind regime that modulates temperature distribution over the area.

As can be seen in Figure 1, the IMS problem is a multidisciplinary problem related to various socio-ecological factors that can affect the economy and society in various ways, through their impact on human, animal health, and various services. These impacts can generate certain economic costs related to control strategies, public health measures, health treatments, productivity losses, information and awareness campaigns, losses in tourism, and other sectors. Economic impacts can be direct or indirect. Direct economic impacts occur when invasive species cause damage that result in increasing costs of various types and can be described as the net increase in spending as a result of the appearance of IMS. These types of economic impacts are those most often clearly defined, as they can be explicitly expressed in monetary values. Control and surveillance programs, private expenditures, and direct medical costs are among the most common categories of direct economic impacts of alien species. Indirect socio-economic effects mainly associated with the introduction of alien pests include, among others, effects on the quality of life of residents, effects on public health, costs associated with new research and management services (for both public and private sectors of the economy), effects on tourism, etc. The complexity of the issue at hand poses challenging questions both for citizens and policy makers with regards to the confrontation of the problem, especially under a turbulent socio-ecological context apparent in South Europe.

Studies conducted in Europe and in USA have examined the socioeconomic benefits and costs associated with the overall mosquito problem [8,9,10,11,12,13]. Most of these studies conclude that the perceived benefits that arise from the reduction of nuisance levels and health threats exceed the costs of prevention and control strategies against various mosquito species, while similar conclusions have been drawn by two preceding studies conducted in Greece [14,15]. The present study aims to enrich the existing literature by investigating: (a) The impact and presence of the invasive mosquito species in Greece at a national and urban level, (b) citizens’ awareness of the *Ae. albopictus* problem, (c) the associated costs and the perceived risks, and (d) the priorities and policy aspects of mosquito control under a turbulent socio-ecological context. Findings are based on the results of a web-based questionnaire [16] as well as on the production of country-level maps produced through official samples (specimens) examined by Benakeion Phytopathological Institute (BPI) offering a dual spatial analysis at a national and regional/metropolitan level. 

## 2. Materials and Methods

### 2.1. Implementation of a Web Survey

The implementation of a web questionnaire follows a process of surveys and evaluations [14,15] aiming to elicit citizens’ preferences for mosquito control strategies as well as to evaluate the effectiveness level of prevention programs in Greece. The questionnaire has been specifically designed to elicit citizens’ opinions for certain socio-economic aspects of the mosquito problem. The overall aim was to examine and then to validate at the national level a set of parameters related to the private prevention costs for IMS and to individual preferences between various mosquito control programs. For this reason, collaboration was established with a web meteorological platform of high visiting frequency (www.meteo.gr) in order to increase the geographical dispersion of the sample. It should be noted that the specific web platform had already implemented a real time monitoring application for the identification of mosquito presence, covering the whole Greek territory. 

The questionnaire was distributed through a popular meteorological data website (www.meteo.gr) with a high number of daily visitors. For the purpose of our survey, a special banner appeared on the home page, from which visitors followed a link to the web survey. The banner appeared randomly to visitors, but a selection bias could arise due to (i) the non-representative nature of the internet population and (ii) self-selection of participants (also called the ‘volunteer effect’ [17], which was possibly related to their interest in mosquito control. The survey took place between September and October 2016 with a total of 1204 responses from all over the country. The final sample follows the regional distribution presented in Table 1. This distribution is quite representative of the population (see Table 1) but it is also a first indicator of regional differences in people’s attitudes and experience of mosquito-associated problems.

The questionnaire contained, first of all, an information form explaining the purpose of the study and general information about the *Ae. albopictus* (including its health risks). The first questions focused on respondents’ knowledge of the *Ae. albopictus*. The following questions were about: (a) The current perceived level of nuisance during day-time as well as during night-time using a numerical scaling score from 1 to 5, where 1 equals “no nuisance” while 5 equals “intolerable nuisance”, also known as the Likert scale [19]; (b) the portion of the year (months) with significant mosquito nuisance; (c) the monthly household expenditure for private prevention measures; as well as (d) the main reasons for taking individual prevention measures (i.e., they had to choose between health risk reduction and nuisance reduction). Then, participants were asked about the importance of taking further public measures for mosquito control (using a 5-point Likert scale). Further questions were then included to identify the main targets of future public control measures/programs. The final section of the questionnaire focused on participants’ demographics (age, residence area, family status).

### 2.2. Production of National Maps on the Distribution of Aedes albopictus Distribution in Greece

Due to geographic position and climatic conditions, Greece is considered suitable for the invasion and establishment of IMS [20,21]. In 2018, the announcement of *Ae. albopictus* detection in Athens resulted in people and stakeholder awareness, and many specimens of tiger-like mosquitoes have been sent to BPI for identification and suggestions for its management. Since 2012, the LIFE CONOPS [22] entomological surveillance started the collection of specimens of tiger-like mosquitoes that were sent to BPI by official authorities, pest control companies, and residents indicating the stakeholder and citizen awareness and nuisance from this “aggressive mosquito species during the daylight” [5]. As also shown in Figure 1, mosquitoes of the *Culex* species are more active during night-time and are associated with the transmission of WNV, while *Ae. albopictus* are known for their intense activity during day-time hours and are associated with the transmission of CHIK and DENV [4,5,15].

Records of *Ae. albopictus* were stored in a geodatabase, which included spatial information about the presence of *Ae. albopictus* per regional unit. The produced thematic maps illustrate the presence of *Ae. albopictus* per regional union for discrete time periods from March 2016 to April 2019.

## 3. Results

### 3.1. Result of the Web Survey at A National Level

According to the results of the web questionnaire, most of the respondents (89.5%) have prior (to the survey) knowledge of the *Ae. albopictus* and to its health risks. It is interesting to note that about 66% of the respondents believe/know that the *Ae. albopictus* is established in their residence area. Regional differences in this response are relatively small (ranging from 55% to 71%) and are not significantly correlated to the actual detection of this mosquito species over Greece until 2016 [23]. Therefore, even though public perception consists of a significant factor concerning the control of *Aedes* species, within the frames of the current study it cannot be used as a safe indicator for monitoring the presence of *Ae. albopictus* in a region/area.

In contrast to a relevant recent study [15] that reported a relatively higher nuisance during the night hours (for the region of Attica), we found that at the national level, night nuisance levels are almost identical with the day-time levels, with a mean value of 3.6 on the 5-point Likert scale (indicating a nuisance level between average and high). Figure 2 presents the distribution of the perceived nuisance level during the night (following the individual responses), as well as the spatial (regional) variation of the mean nuisance value. On the other hand, Figure 3 presents the perceived nuisance levels during the day-time, with a mean value of about 3.6 on the 5-point Likert scale, assumed to be associated with the relative nuisance caused by the *Ae. Albopictus,* which, unlike other mosquito species, appears to be active during the day time. The similarity of recorded levels of perceived nuisance between day and night hours is an indication of the intensification of the *Ae. albopictus* problem in most parts of the country.

According to these results, it can be concluded that respondents living in the regions of Eastern Macedonia and Thrace, Peloponnese, Central Greece, and Western Greece experience a higher day-time biting nuisance that can be attributed to the presence of the *Ae. albopictus*.

Concerning the private (individual) prevention costs, it was found that households are paying, on average, about 17.6 € per month when mosquitoes are active. This estimate is much higher than the other estimates of a similar study [15] for the case of the Attica Region (6.6 €/month). This difference may be attributed to the self-selection of participants, which is likely to be related to their interest in mosquito control, which in turn may depend on the nuisance level. Therefore, these results are likely to be overestimated, but can be used in order to explore the regional variation with regard to prevention costs. In order to do so, we estimated the annual prevention costs by multiplying the monthly costs by the nuisance period. The average annual cost of our sample was found equal to 100.1 €/household (Figure 4). Significant spatial variations were observed in these estimates (Figure 4), as values (annual costs) range from below 80 € in some regions (e.g., Thessaly and the North Aegean) to over 125 € in others (e.g., Eastern Macedonia and Thrace, and Central Greece). This variation may be an indirect indicator of the magnitude of the mosquito problem, which is strongly associated with the nuisance conditions in each area. 

What is more, the web-survey attempted to gather information regarding the preferences of individuals for the diverse mosquito control programs, and particularly about the importance of taking further public measures for mosquito control, as well as about the main targets of future public control measures/programs. In general, about 83% of the survey respondents believe that the actual prevention/control measures are insufficient or inadequate in order to deal with the mosquito problems and therefore, further measures should be taken. Concerning the main targets of these measures, as depicted in Table 2, health impacts were considered as more important than nuisance impacts, validating the previous surveys held in Greece [14,15]. Furthermore, as in the other two studies, diseases from invasive species were considered to be a serious threat. On the other hand, nuisance level and financial burden on households for mosquito control programs were also highly rated, thus constituting important decision factors.

Finally, an important finding of this survey was that citizens seem to be aware of the environmental consequences of mosquito control measures. In particular, about 74% of the sample stated their disagreement with measures that may potentially affect the physical environment and the ecosystems. 

### 3.2. Result of the Web Survey at a Metropolitan Level (Athens)

The analysis of web survey results at the level of the Metropolitan Area of Athens does not differentiate significantly from the findings at a national level. It should be noted that the current analysis spatially corresponds only to those areas for which answers were received within the metropolitan level of Athens. Specifically, Figure 5 and Figure 6 present the perceived levels of nuisance during night and day time. It is found that nuisance during night hours is more intense, indicating a stronger perceived nuisance from *Culex* species. In addition, for certain areas, there appears to be a strong nuisance for both species. However, the intensification of the *Aedes* problem from 2016 to 2019, which will be presented in the next section, could have significantly altered the perceived nuisance and private expenses’ levels.

With regards to the annual prevention costs (Figure 7), these appear to be slightly lower than the country’s average, while high fluctuations exist from area to area also due to certain characteristics such as the presence of parks and cemeteries. What is more, as can be seen in Table 3, there seems to be a higher correlation of prevention costs in relation to perceived nuisance during the day both at a National and Metropolitan level. This is justified by all the above findings, however, the increase in the presence of the Asia tiger mosquito within the whole country level could result in increased expenses for both species.

### 3.3. Results on the Distribution of Aedes albopictus in Greece 

The data provided in Figure 8 aim to present the invasion progress as recorded by a recent study [24], which was gradually updated with data from the surveillance conducted within the framework of LIFE CONOPS project [23]. The thematic maps presented in LIFE CONOPS website distinguish data received from private pest control companies (PCCs) engaged in mosquito management in Greece and data from official samples sent to Benaki Phytopathological Institute and the National School of Public Health.

Based on the results of the current survey, the higher day-time biting in Eastern Macedonia and Thrace, Peloponnese, Central Greece, and Western Greece is in accordance with the results of the official samples (specimens) examined by BPI. In Eastern Macedonia and Thrace, with the exception of Evros prefecture where *Ae. albopictus* was not found yet, a high level of nuisance is confirmed by the positive official specimens. What is more, in Eastern Macedonia and Thrace, until 2016, the information about the presence of *Ae. albopictus* was based on information from pest control companies and citizens. However, after 2017, the presence was confirmed by the samples of LIFE CONOPS oviposition surveillance network. The same is true for Central Macedonia, Peloponnese, and Western Greece. 

## 4. Discussion

The present paper aims to provide an overview of citizens’ perceptions and attitudes towards the problem of invasive mosquitoes, while at the same time presenting an update of the distribution of *Ae. albopictus* as depicted by national surveillance maps. The collection of data was mined through two principal sources, (a) a web-based survey designed and implemented at the national level in Greece and (b) production of surveillance maps. In accordance to the findings of the web survey, the thematic maps present a gradual change in the presence of *Ae. albopictus* as well as their establishment in regional units not previously recorded. In most cases, information given by citizens’ and/or pest control company is usually followed by an official sample, which certifies the community’s perception about *Ae. albopictus* presence. That may imply both a continuation of the citizens’ awareness and the rise of relevant prevention costs incurred at a private level by Greek households. The results of this survey show that nuisance from mosquitoes, though with some regional differences, is significant all over the country, indicating intensification trends. Citizens appear to highly prioritize future policy actions for both invasive and native species while the cost of individual prevention measures was estimated to be quite high (about 100 €/household/year), which can also be a result of the selection bias (i.e., the volunteer effect) due to the nature of the survey (web-survey). However, this variation may be an indirect indicator of the magnitude of the mosquito problem, which is strongly associated with the nuisance conditions in each area. Besides, this revealed behavior concerning prevention can be used as a proxy of individuals’ potential benefits from future improved control programs in each region including several other *Aedes* vector control activities such as an emergency control measures following VBDs imported cases detection and door-to-door control measures in private areas [24,25,26].

Another important outcome of this study is that people all over the country seem to have a higher preference for improved programs targeted at the aversion of health risks over nuisance reduction. The fact that climate change trends may worsen the mosquito problem and increase the risks of new diseases transmission (e.g., Zika virus) is likely to provide even higher preferences/motivations for implementing more efficient mosquito control management plans in the upcoming years [20,21,27]. The evaluation of the socioeconomic costs of invasive mosquitoes is a highly challenging task made even more complex by changing climatic conditions, as well as by globalization and urbanization trends. Taking into account the complexity of the ecological, socioeconomic, and biological conditions, a multi-disciplinary and more holistic approach is needed in order to evaluate the effectiveness of the incurred expenses in improving public health and social welfare, yet at the same time ensuring an ecosystemic equilibrium. 

According to a recent WHO report [1], a synthesis of various parameters such as urban planning, housing, water and sanitation, and insecticide usage should be studied in order to achieve a more holistic estimation of the problem at hand along with the other societal challenges such as unplanned urbanization, financial crisis, and migration. In the case of *Aedes* species control, which requires an intense community participation, space and information should be provided for various societal groups in order to participate in mutual decision-making and action through the implementation of integrated sustainability approaches [28]. In Greece, evaluation of education campaigns and community participation was made for the first time in 2017 [26] and the results strongly suggested that only a visitor inspecting door-to-door habitats in their properties could be enough to stimulate practices towards mosquito breeding sites reduction. 

It should be noted that the pattern and extent of incidence of particular infectious diseases depends, among others, on land-use change, disease-specific transmission dynamics, socio-cultural changes [29,30,31], climate change, and the susceptibility of human populations [32,33]. Based on this fact, it would be rational to explore a synthesis of policies and decisions by including all relevant social groups in the decision-making process. As also highlighted through the findings of the web survey, citizens are aware of the environmental consequences accompanied with mosquito control, even if they cannot be certain about the exact repercussions. Therefore, an informed framework of the socioeconomic cost and benefits of disease regulation programmes should also evaluate the impact of these programmes in ecosystems’ functions, so that both citizens and stakeholders may be able to prioritize different objectives towards the achievement of an ecosystemic equilibrium [28].

## Figures and Tables

**Figure 1 tropicalmed-04-00083-f001:**
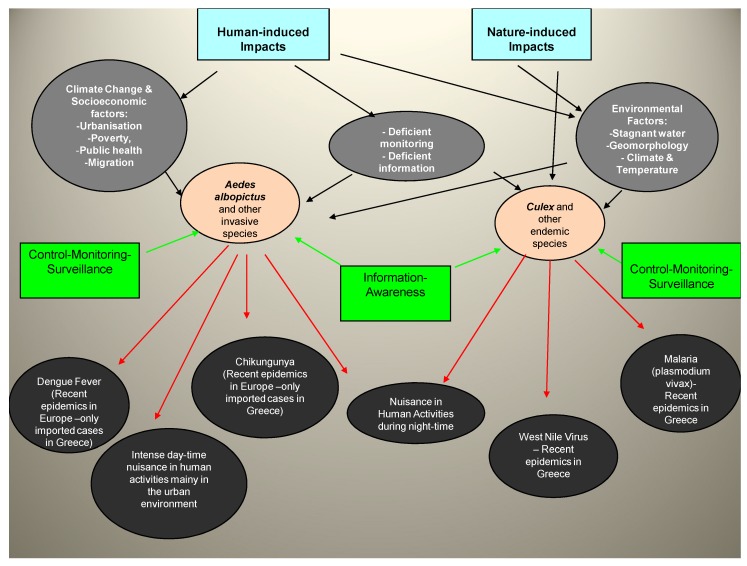
The invasive mosquito species and endemic mosquitoes impact model.

**Figure 2 tropicalmed-04-00083-f002:**
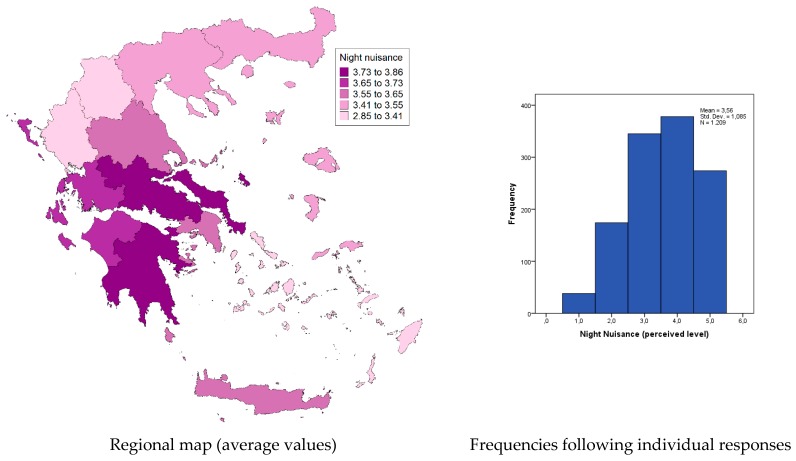
Night nuisance (Likert scale 1–5: 1, no nuisance; 5, intolerable nuisance).

**Figure 3 tropicalmed-04-00083-f003:**
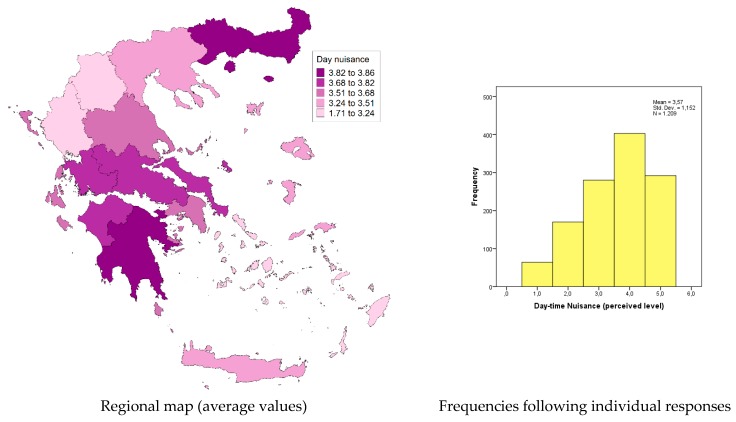
Day nuisance (Likert scale 1–5: 1, no nuisance; 5, intolerable nuisance).

**Figure 4 tropicalmed-04-00083-f004:**
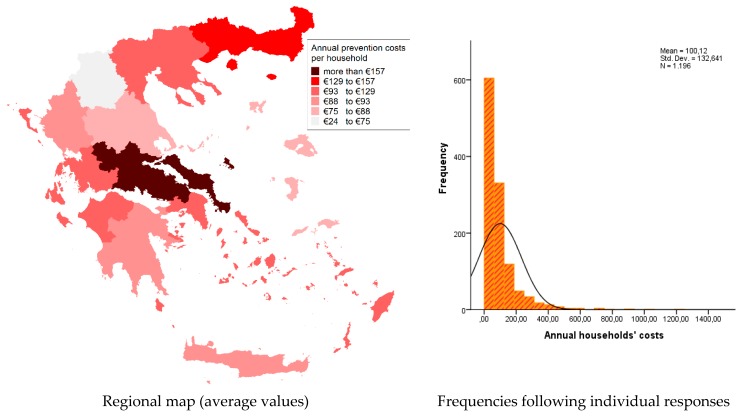
Annual prevention costs (€/year/household).

**Figure 5 tropicalmed-04-00083-f005:**
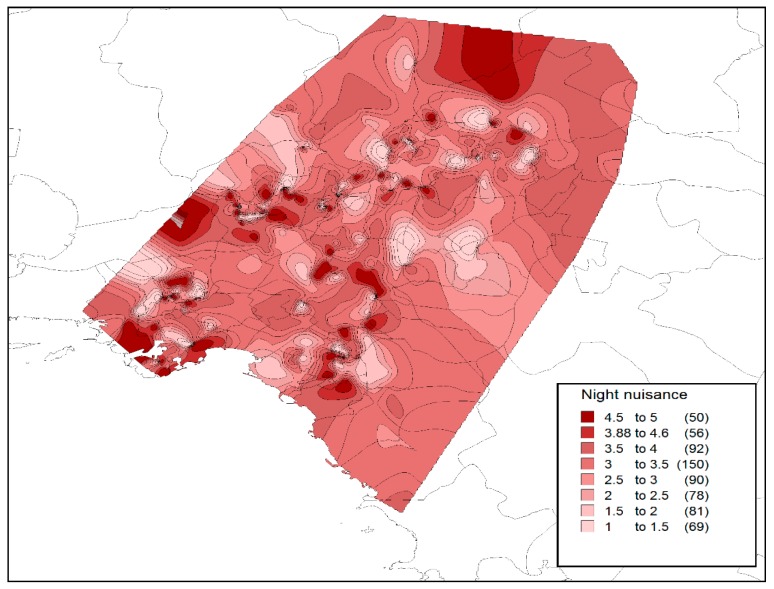
Night nuisance in Athens metropolitan area (Likert scale 1–5: 1, no nuisance; 5, intolerable nuisance).

**Figure 6 tropicalmed-04-00083-f006:**
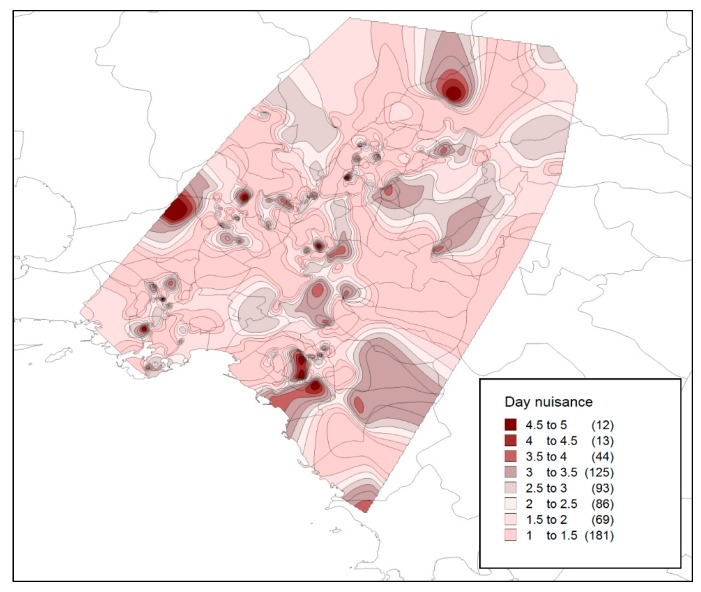
Day nuisance in Athens metropolitan area (Likert scale 1–5: 1, no nuisance; 5, intolerable nuisance).

**Figure 7 tropicalmed-04-00083-f007:**
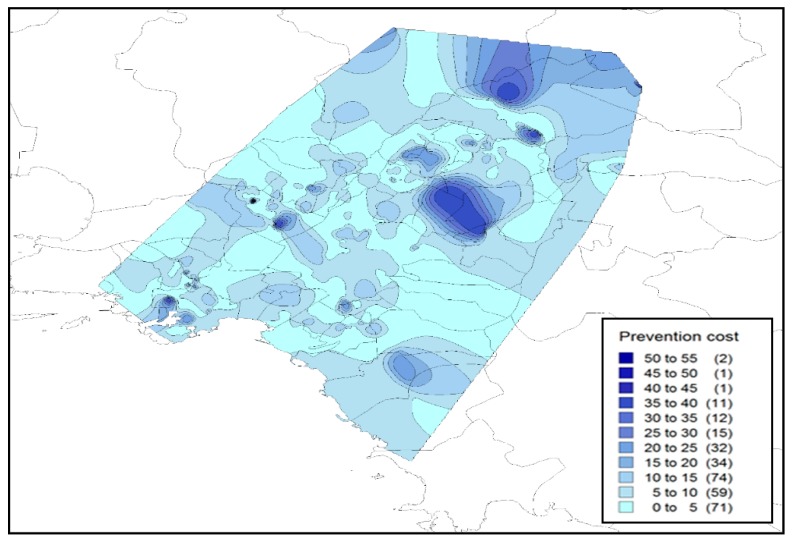
Annual prevention costs in Athens metropolitan area (€/year/household).

**Figure 8 tropicalmed-04-00083-f008:**
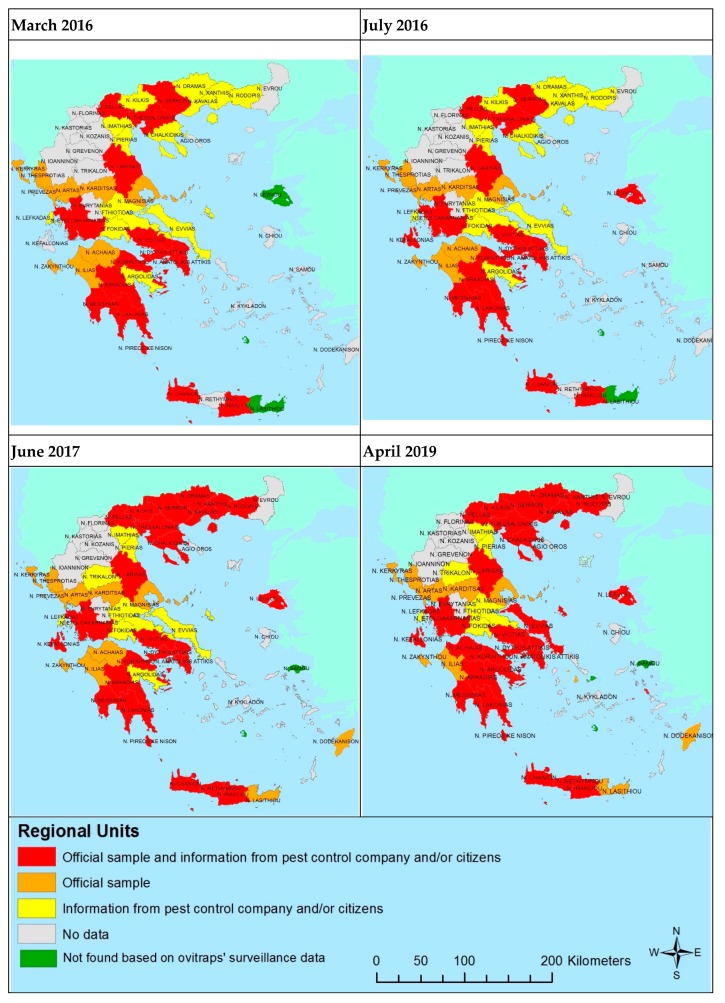
Distribution of *Aedes albopictus* in Greece (2016–2019).

**Table 1 tropicalmed-04-00083-t001:** Sample distribution per region.

	Sample Frequency Percent		Population ^1^ Residents Percent	
Attica	664	55.1%	3,827,624	35.39%
Central Greece	43	3.6%	547,390	5.06%
Central Macedonia	131	10.9%	1,881,869	17.40%
Crete	57	4.7%	623,065	5.76%
Eastern Macedonia and Thrace	49	4.1%	608,182	5.62%
Epirus	35	2.9%	336,856	3.11%
Ionian Islands	33	2.7%	207,855	1.92%
North Aegean	12	1.0%	199,231	1.84%
Peloponnese	49	4.1%	577,903	5.34%
South Aegean	26	2.2%	308,975	2.86%
Thessaly	60	5.0%	732,762	6.78%
Western Greece	38	3.1%	679,796	6.29%
Western Macedonia	7	0.6%	283,689	2.62%

^1^ Data from Population Census in Greece [18].

**Table 2 tropicalmed-04-00083-t002:** Individuals’ rating of the objectives of mosquito control programs (web-survey results).

	Reduction of Mosquito-Borne Disease Risks		Reduction of Nuisance		Cost to Households
	From Native Species ^1^	From Invasive Species ^2^	From Native Species ^3^	From Invasive Species ^4^	From Future Control Programs
Highly important	73.2%	76.7%	47.1%	39.5%	26.8%
Important	19.1%	15.9%	32.3%	25.3%	17.8%
Neutral	5.4%	5.6%	15.7%	20.2%	26.5%
Less important	1.6%	1.2%	4.0%	10.3%	17.4%
Non important	0.7%	0.6%	0.9%	4.7%	11.6%

^1^ for example: malaria, WNV; ^2^ for example: CHIK, DENV, Zika Virus; ^3^ Night nuisance; ^4^ Day-time nuisance.

**Table 3 tropicalmed-04-00083-t003:** Correlation of prevention costs and nuisance levels.

**a. National level**		
	Day Nuisance	Night Nuisance
Prevention costs	0.209	0.222
Day Nuisance		0.466
**b. Athens’ metropolitan area (spatial correlation)**		
	Day Nuisance	Night Nuisance
Prevention costs	0.279	0.305
Day Nuisance		0.505

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
