# Peer review of "Exploring Public Preferences, Priorities, and Policy Perspectives for Controlling Invasive Mosquito Species in Greece"

_tropicalmed, 2019, doi:10.3390/tropicalmed4020083_

Round 1
Reviewer 1 Report
This paper is interesting and topical, but very wide ranging and full of information which is not always clearly explained. It falls into three parts - the web survey on risk perceptions, economic impacts, and mosquito distribution maps of Aedes albopictus and changes over time between 2016 and 2019. The questionnaire results are interesting and novel.These three parts do not hang together very well (the geographic areas used for the questionnaire results and the mosquito surveillance are not the same) and there is little formal comparison between the results in different sections.
It would be important to include statistical comparison between the perceived nuisance scale in Figure 2 and 3, since there are conclusions made from this difference.
Changes between March 2016 to April 2019 are described as 'constant intensification' but that seems overinterpretation from the map 1. I see more limited and gradual change,plus the data source changes over time. The paragraph in lines 236 to 239 is very speculative given that the web survey was done only in a cross-sectional way and not over time so it's not clear that there is a rise in prevention costs over time.. This section seems to belong in Discussion not Results.
In line 246 it stated that there is comparison of citizens' perception with surveillance records, but this needs clearer justification. The only formal comparison is of nuisance level and preventoin costs (Table 3).
The Discussion part around Table 4 and the last paragraph does not seem to arise from or relate well to the findings of the paper and is speculative. The link between mosquito control, effectiveness of control method and human health is a bit over-emphasized since there is nothing about human health in the paper. (just perceptions, prevention methods used and mosquito distribution). I believe the Discussion could be shortened and made more focused.
Minor points:
Figure 1: It is surprising that day-time nuisance is not included in the diagram because that is a well known effect of Ae albopictus
Line 74: Refs 6 to 13 are cited but it should be noted that 12 and 13 were actually done in Greece so could be mentioned separately as they are precursors to the current study.
Lines 168 to 170. This sentence is not clear.
Please reorder Figure 5 and 6 so that order of night and day stays consistent with earlier figures and the text.
Table 3 needs footnotes to explain the asterisks.
Line 260 talks about 'health aversion' but I think you mean 'health promotion' or 'health problem aversion'.
Paragraph from line 270 to 275 - a very long sentence and mixing up recommendations for study and suggestions for action. It should be broken up and also related more to the findings in the paper, as should the rest of this paragraph.
line 277 - it is not really a problem of mosquito physiology, but more likely organism's bionomics.
Author Response
This paper is interesting and topical, but very wide ranging and full of information which is not always clearly explained. It falls into three parts - the web survey on risk perceptions, economic impacts, and mosquito distribution maps of Aedes albopictus and changes over time between 2016 and 2019. The questionnaire results are interesting and novel. These three parts do not hang together very well (the geographic areas used for the questionnaire results and the mosquito surveillance are not the same) and there is little formal comparison between the results in different sections.
Dear Reviewer,
We appreciate your interest in our manuscript as well as your constructive criticism. In this revised version we have made a systematic effort to harmonize the three parts of the study by highlighting their district importance. Our overall aim has been to highlight the findings of the web survey conducted along with certain socioeconomic conclusions drawn from it, with recent surveillance data on the presence of Ae. albopictus during the last years in Greece and to highlight certain policy challenges related to the control of the Asian tiger mosquito in Greece.
It would be important to include statistical comparison between the perceived nuisance scale in Figure 2 and 3, since there are conclusions made from this difference.
We have tried to highlight the statistical comparison and relevant conclusions in lines 186-190.
Changes between March 2016 to April 2019 are described as 'constant intensification' but that seems overinterpretation from the map 1. I see more limited and gradual change, plus the data source changes over time. The paragraph in lines 236 to 239 is very speculative given that the web survey was done only in a cross-sectional way and not over time so it's not clear that there is a rise in prevention costs over time.. This section seems to belong in Discussion not Results.
Following the reviewer’s comment, (previous) lines 236 to 239 were modified to give a better interpretation of the map. Furthermore, the modified text was transferred in the Discussion section (lines 398-404).
In line 246 it stated that there is comparison of citizens' perception with surveillance records, but this needs clearer justification. The only formal comparison is of nuisance level and preventoin costs (Table 3).
In line with the above comment, and for reason of content's clarity lines 398 to 404 have been reformulated.
The Discussion part around Table 4 and the last paragraph does not seem to arise from or relate well to the findings of the paper and is speculative. The link between mosquito control, effectiveness of control method and human health is a bit over-emphasized since there is nothing about human health in the paper. (just perceptions, prevention methods used and mosquito distribution). I believe the Discussion could be shortened and made more focused.
We thank you for your comment. The whole discussion section has been reformulated and for reasons of cohesion with the rest of the content, the major part of the last paragraph has been deleted.
Minor points:
Figure 1: It is surprising that day-time nuisance is not included in the diagram because that is a well known effect of Ae albopictus
A specific part of the figure has been restructured in order to highlight the day-time nuisance of Ae. albopictus.
Line 74: Refs 6 to 13 are cited but it should be noted that 12 and 13 were actually done in Greece so could be mentioned separately as they are precursors to the current study.
The two preceding Greek studies were mentioned separately within manuscript.
Lines 168 to 170. This sentence is not clear.
Sentence has been deleted.
Please reorder Figure 5 and 6 so that order of night and day stays consistent with earlier figures and the text.
A reorder of Figures has been done according to the comment.
Table 3 needs footnotes to explain the asterisks.
Asterisks have been deleted.
Line 260 talks about 'health aversion' but I think you mean 'health promotion' or 'health problem aversion'.
Thank you very much for the comment, sentence has been corrected.
Paragraph from line 270 to 275 - a very long sentence and mixing up recommendations for study and suggestions for action. It should be broken up and also related more to the findings in the paper, as should the rest of this paragraph.
In line to a previous comment on the general restructuring of the discussion section, those particular lines have been corrected in order to improve overall content's clarity.
line 277 - it is not really a problem of mosquito physiology, but more likely organism's bionomics.
Line 277 has been deleted along with major part of the last paragraph.
Reviewer 2 Report
I have reviewed the manuscript titled “Exploring public preferences, priorities and policy perspectives for controlling invasive mosquito species in Greece” by Kolimenakis et al.
The manuscript reports on a study of public perceptions, as measured through an online survey, of exotic mosquito threats. With reference to changes in exotic mosquito detections in surveillance programs, the results of public attitudes are discussed with reference to the policy response of local authorities.
Given the critical importance of community involvement in the management of exotic mosquito threats in many parts of the world, studies of this nature can make a valuable contribution to the preparation of strategic responses to mosquito threats. They can also assist researchers looking to tap into community concerns to inform citizen science based approaches to future mosquito studies.
The paper is generally well written, methods sounds and appropriate presentation and analysis of results included.
I do not have any major comments on the study. However, there are some suggestions that could improve the manuscript overall.
I would like to see some discussion on the justification of splitting survey across the different regions. For those unfamiliar with Greece, a brief description of any climatic, environmental, or entomological aspects that differentiate the regions. Similarly, it would be useful to include an annotated map highlighting these regions.
The authors should provide more information on the differences between nuisance impacts associated with Aedes albopictus compared to other endemic/exotic mosquitoes in this region of Europe. As I understand it, Culex pipiens group mosquitoes can cause nuisance, especially indoors at night. While the authors do touch on this, I think it is important to clearly articulate that there are differences between the severity of biting, but also general indoor/outdoor nuisance caused by these different mosquitoes. Further discussion required to assist understanding community attitudes and their difference in response to the different mosquitoes (either geographically or at different points in the day).
Throughout the manuscript, the authors vary in their use of either scientific (Ae. albopictus) or common name (Asian tiger mosquito) of mosquitoes. For consistency, one or the other should be used.
Minor comments
Line 46. Given the authors are discussing the viruses, change “Chikungunya” to “chikungunya” and “Dengue” to “dengue”. It would also be useful to include here the abbreviations for the viruses, DENV, CHIKV, and WNV (similarly, see footnote to Table 3)
Figure 1. The colour scheme used makes it very difficult to read the text inside the red boxes. I would actually prefer the figure to be black and white only but I also appreciate that colour can assist in differentiating the different categories. I would also avoid using the acronym IMS in the figure caption, write out in full.
Line 121. Remove itallics from scientific name in subheading
Line 132. Change “Albopictus” to “albopictus”
Line 142. Abbreviation required, change “Aedes albopictus” to “Ae. albopictus” (Note, there are a number of other occasions throughout manuscript where abbreviation can be made, see lines 230, 232, 244 as examples but manuscript should be reviewed elsewhere)
Line 113/145. I am not familiar with the “Likert scale”, the authors should include a brief description in methods
Line 215. Change “Asia tiger” to “Asian tiger” (but also see note regarding use of scientific and common names
Table 3. Can formatting be modified to ensure table is not split across two pages?
Line 220. Remove itallics from scientific name in subheading and change “Albopictus” to “albopictus”
Figure 8/Map 1. These figures are very difficult to read due to the colour and small size of text, is it possible to increase the size of each map within the figure box. A more detailed figure legend is required, and I assume this should be referred to as Figure 8?
Line 279. Italicise “Aedes”
References: All references should be reviewed to ensure that they meet formatting requirements of the journal, I note that in a number of references, scientific names in the title of manuscripts cited are not itallised.
Author Response
Dear Reviewer,
We would like to thank you very much for your positive comments and relevant suggestions to improve our submitted manuscript.
I would like to see some discussion on the justification of splitting survey across the different regions. For those unfamiliar with Greece, a brief description of any climatic, environmental, or entomological aspects that differentiate the regions. Similarly, it would be useful to include an annotated map highlighting these regions.
Relevant climatic and meteorological data were added in lines 56-65, while an updated version of the surveillance maps can be found in lines 347-348 and their discussion in lines 394-400.
The authors should provide more information on the differences between nuisance impacts associated with Aedes albopictus compared to other endemic/exotic mosquitoes in this region of Europe. As I understand it, Culex pipiens group mosquitoes can cause nuisance, especially indoors at night. While the authors do touch on this, I think it is important to clearly articulate that there are differences between the severity of biting, but also general indoor/outdoor nuisance caused by these different mosquitoes. Further discussion required to assist understanding community attitudes and their difference in response to the different mosquitoes (either geographically or at different points in the day).
The community aspects were addressed in lines 432-435. In addition, some information on the Culex and Aedes nuisance activities were added in lines 163-166 of the revised manuscript.
Throughout the manuscript, the authors vary in their use of either scientific (Ae. albopictus) or common name (Asian tiger mosquito) of mosquitoes. For consistency, one or the other should be used.
Thank you very much for the comment. We decided to use the scientific name Ae. albopictus throughout the whole revised manuscript.
Minor comments
Line 46. Given the authors are discussing the viruses, change “Chikungunya” to “chikungunya” and “Dengue” to “dengue”. It would also be useful to include here the abbreviations for the viruses, DENV, CHIKV, and WNV (similarly, see footnote to Table 3)
A correction was made according to comment.
Figure 1. The colour scheme used makes it very difficult to read the text inside the red boxes. I would actually prefer the figure to be black and white only but I also appreciate that colour can assist in differentiating the different categories. I would also avoid using the acronym IMS in the figure caption, write out in full.
A certain editing has been done to the figure's formatting.
Line 121. Remove itallics from scientific name in subheading
A correction was made according to comment.
Line 132. Change “Albopictus” to “albopictus”
Corrections were made according to comment.
Line 142. Abbreviation required, change “Aedes albopictus” to “Ae. albopictus” (Note, there are a number of other occasions throughout manuscript where abbreviation can be made, see lines 230, 232, 244 as examples but manuscript should be reviewed elsewhere)
A correction was made according to comment.
Line 113/145. I am not familiar with the “Likert scale”, the authors should include a brief description in methods
A brief description of the ordinal numerical scaling system known as "Likert Scale" has been added in lines 136-138.
Line 215. Change “Asia tiger” to “Asian tiger” (but also see note regarding use of scientific and common names
Corrections were made according to comment.
Table 3. Can formatting be modified to ensure table is not split across two pages?
Certain changes were made to ensure tables and figures are not split across two pages.
Line 220. Remove itallics from scientific name in subheading and change “Albopictus” to “albopictus”
Corrections were made according to comment.
Figure 8/Map 1. These figures are very difficult to read due to the colour and small size of text, is it possible to increase the size of each map within the figure box. A more detailed figure legend is required, and I assume this should be referred to as Figure 8?
The figures were modified according to the above mentioned comment and “Map 1” was changed to “Figure8”.
Line 279. Italicise “Aedes”
A correction was made according to comment.
References: All references should be reviewed to ensure that they meet formatting requirements of the journal, I note that in a number of references, scientific names in the title of manuscripts cited are not itallised.
References were reviewed in order to meet the formatting requirements of the journal.